# Train Offline, Test Online: A Real Robot Learning Benchmark

**Gaoyue Zhou\***[1]    **Victoria Dean\***[1]    **Mohan Kumar Srirama**[1]

**Aravind Rajeswaran**[2,5]    **Jyothish Pari**[3]    **Kyle Hatch**[4]    **Aryan Jain**[5]    **Tianhe Yu**[4]

**Pieter Abbeel**[5]    **Lerrel Pinto**[3]    **Chelsea Finn**[4]    **Abhinav Gupta**[1]

[1]Carnegie Mellon University    [2]University of Washington    [3]New York University

[4]Stanford University    [5]University of California, Berkeley

## Abstract

Three challenges limit the progress of robot learning research: robots are expensive (few labs can participate), everyone uses different robots (findings do not generalize across labs), and we lack internet-scale robotics data. We take on these challenges via a new benchmark: Train Offline, Test Online (TOTO). TOTO provides remote users with access to shared robots for evaluating methods on common tasks and an open-source dataset of these tasks for offline training. Its manipulation task suite requires challenging generalization to unseen objects, positions, and lighting. We present initial results on TOTO comparing five pretrained visual representations and four offline policy learning baselines, remotely contributed by five institutions. The real promise of TOTO, however, lies in the future: we release the benchmark for additional submissions from any user, enabling easy, direct comparison to several methods without the need to obtain hardware or collect data.

## 1  Introduction

One of the biggest drivers of success in machine learning research is arguably the availability of benchmarks. From GLUE [Wang et al., 2018] in natural language processing to ImageNet [Deng et al., 2009] in computer vision, benchmarks have helped identify fundamental advances in many areas. On the other hand, robotics as a field struggles to establish common benchmarks due to the physical nature of evaluation. The experimental conditions, objects of interest, and even hardware varies across labs, often making algorithms sensitive to implementation details. Finally, the difficulties of purchasing, building, and installing hardware and software infrastructure make it challenging for newcomers to contribute to the field.

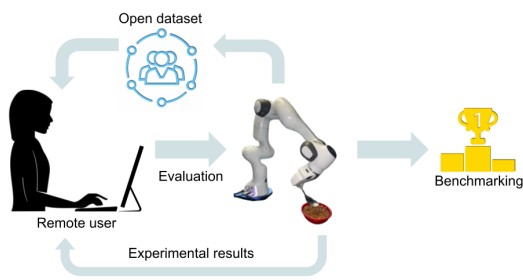

Figure 1: **Train Offline, Test Online**: Our benchmark lets remote users test offline learning methods on shared robots.

Offline Reinforcement Learning Workshop at Neural Information Processing Systems, 2022

For robotics research to advance, we clearly need a common way to evaluate and benchmark different algorithms. A good benchmark will not only be fair to all algorithms but also have low participation barrier: setup to evaluation time should be as low as possible. Efforts like YCB [Calli et al., 2015] and RB2 [Dasari et al., 2021] aim to standardize objects and tasks, but the onus of setting up infrastructure still lies with each lab. A simple way to overcome this is the use of a common physical evaluation site, as the Amazon Picking Challenge [Correll et al., 2016] and DARPA Robotics Challenges [Buehler et al., 2009, Krotkov et al., 2017, Seetharaman et al., 2006] have. However, the barrier is still high since participants must set up their own training infrastructure. Both of the above frameworks leave the method development phase unspecified and struggle to provide apples to apples comparisons.

Many robot learning algorithms do online training, where a policy is learned concurrently with data collection. One way to standardize online training is with simulation [Todorov et al., 2012, Yu et al., 2020, Brockman et al., 2016, Zhu et al., 2020]. While simulation mitigates issues with variation across labs, the findings from simulated benchmarks may not transfer to the real world. On the other hand, if we conduct online training in the real world, comparison across labs becomes difficult due to physical differences. In recent years, larger offline datasets have surfaced in robotics [Dasari et al., 2019, Mandlekar et al., 2018, Collins et al., 2019], and with them the rise of offline training algorithms. From imitation learning to offline RL, these algorithms can be trained using the same data and tested in a common physical setup.

Inspired by this observation, we propose a new robotics benchmark: **TOTO (Train Offline, Test Online)**. TOTO has two key components: (a) a large-scale offline manipulation dataset to train imitation learning and offline RL algorithms; (b) a shared hardware setup where users can evaluate their methods now and going forward. Because all participants train using the same publicly-released dataset and evaluate on shared hardware, the benchmark provides a fair apples-apples comparison.

TOTO paves a path forward for robot learning by lowering the entry barrier: when designing a new method, a researcher can train their policy on our dataset, evaluate it on our hardware, and directly compare it to the existing baselines for our benchmark. TOTO means no more time devoted to setting up hardware, collecting data, or tuning baselines for one individual's environment. In this paper, we lay out our benchmark design and present the initial methods contributed by benchmark beta testers across the country. These results show that our benchmark suite is challenging yet possible, providing room for growth as users iterate on TOTO.

## 2   Related Work

For a thorough description of work related to remote robotics benchmarking, we refer to the Robotics Cloud concept paper [Dean et al., 2022]. Here we describe related work specific to our instantiation of a robotics cloud (TOTO).

### 2.1   Shared Tasks and Environments

A necessary step in comparing method performance is evaluation on a common task. Common tasks might mean a standard object set such as YCB [Calli et al., 2015], which can be distributed to remote labs, allowing for shared metrics like grasp success on these objects. The Ranking-Based Robotics Benchmark (RB2) [Dasari et al., 2021] provides four common manipulation tasks (similar to those we use, described in Section 3.2) as well as a framework for comparing and ranking methods across results from multiple labs. Another route is sharing the environment itself, as the Amazon Picking Challenge [Correll et al., 2016] and DARPA Robotics Challenges [Buehler et al., 2009, Krotkov et al., 2017, Seetharaman et al., 2006] have done. Sharing tasks or environments gives metrics by which we can compare approaches. However, users must still develop the approach on their own hardware in their own lab, and recreating identical environment setups is quite challenging.

### 2.2   Shared, Remote Robots

Going one step further, remotely-accessible robots can be shared across the community, enabling method development and evaluation without users acquiring their own hardware. Georgia Tech's Robotarium [Pickem et al., 2017] allows for remote experimentation of multi-agent methods on a physical robotic swarm, which has been extensively used not just in research but also in education. OffWorld Gym [Kumar et al., 2019] provides remote access to navigation tasks using a mobile robot,

with closely mirrored simulated and physical instances of the same environment. A recent survey paper [Sun et al., 2021] provides an overview of robotic grasping and manipulation competitions, including some that involve remotely-accessible, shared robots like [Liu et al., 2021]. Finally, most closely related to our work, the Real Robot Challenge [Funk et al., 2021] runs a tri-finger manipulation competition on cube reorientation tasks. The success of the Real Robot Challenge framework inspires our work, which also allows for evaluation of manipulation tasks on shared robots. Our work, however, is designed to evaluate robot *learning* through challenging variations (lighting, unseen test objects, etc.) and an image-based dataset (as opposed to assuming ground-truth state access).

## 2.3 Open-Source Robotics Datasets

Collecting real-world robotics data is challenging and expensive due to physical constraints like environment resets and hardware failures. Thus open-source datasets serve an important role in the field by enabling larger-scale offline robot learning. Some work has improved the way we collect robotics data, such as self-supervised grasping [Pinto and Gupta, 2016] and further parallelization of robots [Levine et al., 2018]. RoboTurk [Mandlekar et al., 2018] provides a system for simple teleoperated data collection which can be executed remotely. Much work in robot learning has introduced datasets more generally, such as MIME [Sharma et al., 2018] (8260 demonstrations over 20 tasks), RoboNet [Dasari et al., 2019] (162,000 trajectories collected across 7 robots), and Bridge Data (7,200 demonstrations across 10 environments). However, it is hard to understand the value of these datasets without a common evaluation platform, something that Collins et al. [2019] addresses by using simulation to replicate a real-world dataset. In contrast, we address this issue with real-world evaluation that matches the domain of the data collection. Our initial dataset is 2,898 trajectories, but this will grow over time as we add evaluation trajectories collected from users' policies.

## 2.4 Offline Robot Learning

Our benchmark focuses on offline robot learning, including imitation learning and offline RL. Our initial baselines are described and contextualized in Section 5.2.

# 3 The TOTO Benchmark

Our benchmark focuses on manipulation, as it is more challenging than grasping alone but more feasible than navigation or locomotion, which might require more space, safety checks, and manual resets. The robots (Section 3.1) are set in environments that enable a set of manipulation tasks described in Section 3.2. We collect an initial dataset on these tasks, detailed in Section 3.3. Finally, in Section 3.4, we present the evaluation protocol for all policies contributed to our benchmark.

## 3.1 Hardware

Our hardware includes a Franka Emika Panda robot arm and workstation for real-time inference. We use a simple and common joint position control stack that runs at 30 Hz. Actions are specified as joint targets, which are translated into motor control signals using an underlying high-frequency PD controller. We use joint position control because end effector control using X,Y, Z positions alone is not feasible to solve our tasks: for example, the orientation of the gripper must change as the robot pours. We use an Intel D435 RealSense camera for recording RGB-D image observations.

We allow users to opt for a lower control frequency if desired. The training data can be subsampled by taking one of N frames since the actions are in absolute joint angles. We decrease the test time control frequency accordingly.

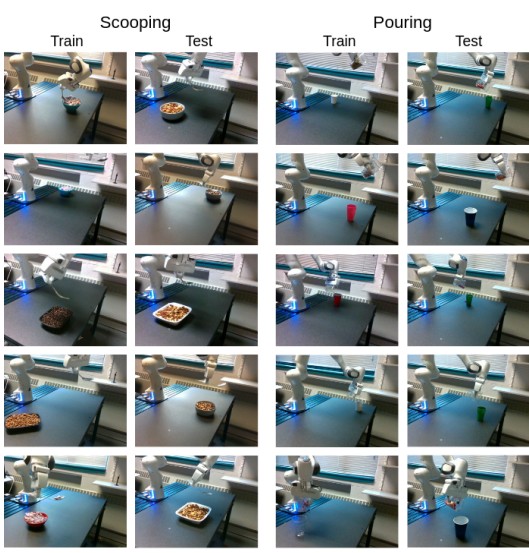

Figure 2: **TOTO Task Suite.** Our benchmark tasks are pouring and scooping. Each involves challenging variations in objects, position, and more.

## 3.2 Tasks

The task suite consists of two household manipulation tasks that humans encounter on a daily basis, similar to those introduced in prior work [Dasari et al., 2021, Bahl et al., 2021]. The tasks are pouring and scooping, excluding the easiest and hardest RB2 tasks (zipping and insertion). Example image observations for these tasks are shown in Fig. 2. To see the original task designs, please refer to RB2: `https://rb2.info`. Our tasks differ from those in RB2 in a few ways. We randomize the robot start state at the beginning of each episode. We apply a bit more noise to the target object locations. We have different combinations of objects based on availability. Lastly, we do not normalize the reward: the reward is the weight in grams of the material successfully scooped or poured. Below we detail per-task specifications.

**Scooping** The training set includes all combinations of three target bowls, three materials, and six target bowl locations (front left, front center, front right, back left, back center, and back right).

**Pouring** The training set includes all combinations of four target cups, two materials, and six target cup locations (same locations as scooping). The cup in the robot gripper is the same in all experiments (clear plastic, enabling better perception of the material remaining in the cup).

## 3.3 Dataset

A key pillar of our benchmark is the release of a manipulation dataset. Dataset statistics (number of trajectories, average trajectory length, success rate, and data collection breakdown) are shown in Table 1. The initial release includes between 1000 and 2000 trajectories per task.

Table 1: Dataset overview

| Task | Trials | Length | Success | Teleop | BC | Replay |
|------|--------|--------|---------|--------|-----|--------|
| Scooping | 1895 | 495 | 0.690 | 41% | 33% | 26% |
| Pouring | 1003 | 324 | 0.977 | 99% | 0% | 1% |

Pouring data collection using replay and behavior cloning proved challenging to reset (unsuccessful trials require more cleanup), so it was nearly all collected with teleoperation. Each recorded trajectory includes RGB-D video, robot actions (joint angle targets), joint states (joint angles), and task metrics (rewards). To improve diversity, the data were collected with three techniques, each described below.

**Teleoperation** We collected the majority of trajectories with teleoperation using Puppet [Kumar and Todorov, 2015]. The human controls the robot in an intuitive end effector space using an HTC Vive virtual reality headset and controller. While this teleoperation is theoretically possible to use remotely, we collect the data with the human and robot in the same room, giving the human direct perception of the scene. Our multiple teleoperators have different dominant hands, leading to more diverse data. Most teleoperation trials are successful.

**Behavior cloning rollouts** After teleoperation trajectories are collected, we train simple, state-based behavior cloning (BC) policies on each target location, so no visual perception is required. We roll out these trajectories with some noise added to actions at each timestep. The amount of noise varies across trajectories for additional diversity.

**Trajectory replay** Finally, we replay individual teleoperated trajectories with added noise. While these might seem overly similar to the original teleoperated trajectories, keep in mind that conditions like lighting also vary with time of day, so this replay still expands the dataset in other ways.

## 3.4 Evaluation Protocol

We evaluate each task in a variety of test settings. We have two unseen objects (bowls and cups) and one unseen material (mixed nuts for scooping and Starburst candies for pouring). We evaluate three object locations seen during training (front left, front center, and front right) and three unseen test locations. We evaluate three training seeds of each method. We initialize the robot with a randomly sampled pose at the beginning of each trajectory. However, the robot's initial poses are kept the same

across seeds to ensure minimal variance. Combining 2 objects, 1 material, 3 locations, and 3 seeds means that each method is evaluated across 18 trials each for train and test locations. We report mean and variance of these 18 trials.

# 4  Benchmark Use

Here we introduce the framework for our benchmark. TOTO is designed to make the user workflow (Section 4.1) easy for newcomers with well-documented software infrastructure (Section 4.2) including examples and tests.

## 4.1  User Workflow

We provide a real-world dataset (Section 3.3) collected using our hardware setup (Section 3.1). Participants optionally use our software starter kit (Section 4.2) and locally train policies of their choosing using this data.

Users submit policies through Google Drive for evaluation on our real-world setup. They do not receive the low-level data from these evaluation trials; they simply receive a reward and high-level video to guide algorithm development, but not enough data to be used effectively for online training.

We run the real-world evaluations while an engineer is present to supervise; thus the evaluation turnaround time is currently around 12 hours (depending on the time of day submitted). Our goal is to place the emphasis on offline learning and prevent overfitting, thus removing the need for real-time results or large quantities of evaluation.

As new users evaluate methods after the paper release, we will post (anonymous) evaluation scores for each attempt on a website leaderboard. We will also periodically add data collected by the users' policies to the original dataset.

## 4.2  Software Infrastructure

Our software starter kit includes documented code and instructions for policy formatting and dataset usage. We have open-sourced baseline code, trajectory data, and pretrained models (see our website). These components ensure that TOTO is easily accessible to a broad portion of the robotics, ML, and even computer vision communities.

We adapt the agent format from Ke et al. [2021], which requires a `predict` function taking in the observation and returning the action. We also use a standard config format and require an `init_agent_from_config` function to create the agent.

We provide users with code for training an example image-based BC agent and a docker environment which wraps the minimum required dependencies to run this code. Users can optionally extend the docker containers with additional dependencies. We also provide a stub environment which users can use to locally evaluate whether the agent's predictions are compatible with our robot environment. This setup allows resolution of all agent format and library dependency issues before users submit their agents for evaluation.

# 5  Baselines

We highlight the importance of establishing a benchmark by running two sets of experiments: (a) what is a good visual representation for manipulation? and (b) what is a good offline algorithm for policy learning? To test the benchmark infrastructure, we have solicited baseline implementations for both experiments from several labs.

## 5.1  Visual Representation Baselines

A core unanswered question, due to the lack of benchmarking, is what is a good visual representation for manipulation? Is ResNet trained on ImageNet great or do self-supervised approaches outperform supervised models? We evaluate five visual representations provided by TOTO users from multiple labs. Two are trained on our data (in-domain) and three are generically pretrained.

`Resnet50` refers to the model trained with supervised learning on ImageNet [He et al., 2015].

`MoCo (Generic)` refers to Momentum Contrast (MoCo) trained on ImageNet [He et al., 2020], while `MoCo (In-Domain)` is trained on our data with crop-only augmentations [Parisi et al., 2022].

`R3M` (Reusable Representations for Robot Manipulation) [Nair et al., 2022] is trained on Ego4D [Grauman et al., 2022] with time-contrastive learning and video-language alignment. `R3M`, `MoCo`, and `Resnet50` use the 2048-dimensional embedding vector following the fifth convolutional layer.

`BYOL` (Bootstrap Your Own Latent) [Grill et al., 2020] is a self-supervised representation learning method trained on our dataset. The BYOL representation embedding size is 512.

These representations performed the best among a larger set of vision models on which we ran an initial brief analysis (including offline visualizations and BC rollouts). Additional representations that performed less well (and therefore are not included as baselines) included CLIP [Radford et al., 2021] and a lower-level MoCo model (from the third layer instead of the fifth).

## 5.2 Policy Learning Baselines

Remote users have contributed the policy learning baselines detailed below. These methods span the spectrum from nearest neighbor querying to BC to offline reinforcement learning (RL). They were selected according to each TOTO contributor's expertise with approach coverage in mind.

All methods use RGB image observations, and some run these images through a frozen, pretrained vision model before passing the resulting embedding to a policy.

`BC` is trained on top of each vision representation baseline. Closed-loop BC predicts a new action every timestep, while open-loop BC predicts a sequence of actions to execute without re-planning. Our BC baseline is *quasi* open-loop: training trajectories are split into 50-step action sequences, and the policy is trained to predict such a sequence given the current observation. During evaluation, these 50 actions are executed between each prediction step. We find that this performs better than closed-loop or open-loop alone: closed-loop struggles without history, and open-loop is challenging with our variable-length tasks. We filter the training data to only include trajectories with nonzero reward [Chen et al., 2021].

`IQL` (Implicit Q-learning) [Kostrikov et al., 2021] uses the open-source implementation from the `d3rlpy` package [Seno and Imai, 2021]. We use `MoCo (In-Domain)` as a frozen visual representation since it performed the best in our comparison of representations with BC. We concatenate the frozen image embeddings with the robot's joint angles as the input state to the model.

`VINN` (Visual Imitation through Nearest Neighbors) [Pari et al., 2021] is a nearest neighbor policy using an image encoder trained with `BYOL` [Grill et al., 2020]. While using nearest neighbors as a policy has been previously explored [Mansimov and Cho, 2018], this approach alone does not scale well to high-dimensional observations like images. BYOL maps the high-dimensional observation space to a low dimension to obtain a robust policy. VINN was originally closed-loop (query and execute a new action at each timestep), but in this work we mirror the 50-step quasi open-loop approach used in the BC baseline (described above).

`DT` (Decision Transformers) [Chen et al., 2021] recasts offline RL as a (conditional) sequence modeling task using transformers. Similar to BC, it is trained to predict the action in the dataset, but conditions on the trajectory history as well a target return (desired level of performance). We use the Hugging Face DT implementation. The model receives an RGB image and the robot's joint angles: the former is embedded using `MoCo (In-Domain)` and concatenated with the latter at each time step. DT uses a sub-sampling period of 8 and a history window of 10 frames. For inference and evaluation, the target return prompt is approximately chosen as the mean return from the top 10% of trajectories in the dataset for each task.

# 6 Experimental Results

## 6.1 Visual Representation Comparison Using BC

Our first experiments compare BC agents using the vision representations detailed in Section 5.1 and evaluated with the protocol described in Section 3.4. The success rates across all representations and tasks are visualized in Fig. 3, and the numerical rewards are presented in Table 2.

Table 2: Performance of vision representations with BC across train and test locations.

| | Model | Scooping | | Pouring | |
|---|---|---|---|---|---|
| | | Reward | Success % | Reward | Success % |
| In Domain | BYOL | 4.39 | 72.2% | 20.22 | 66.6% |
| | MoCo | **7.42** | **83.3%** | **22.86** | **72.2%** |
| Out of Domain | MoCo | 2.11 | 33.3% | 14.89 | 55.5% |
| | ResNet50 | 2.83 | 47.2% | 18.86 | 50.0% |
| | R3M | 2.97 | 44.4% | 6.94 | 33.3% |

These results show that finetuning the MoCo model on our data outperforms the generic version, as expected. `MoCo (In-Domain)` achieves the highest success rate and average reward on both scooping and pouring, followed by `BYOL`, the other in-domain model. In general, the relative performance between models is mostly consistent across scooping and pouring. Resnet50 and `MoCo (Generic)` perform slightly better on pouring than on scooping.

Fig. 3 also visualizes performance differences due to object locations. Locations seen during training perform better, as expected, but performance does not degrade significantly, suggesting that the representations have a generalizable notion of where the target object is. Surprisingly, the two representations trained on our data (`MoCo (In-Domain)` and `BYOL`) perform equally good or even slightly better on unseen locations for scooping.

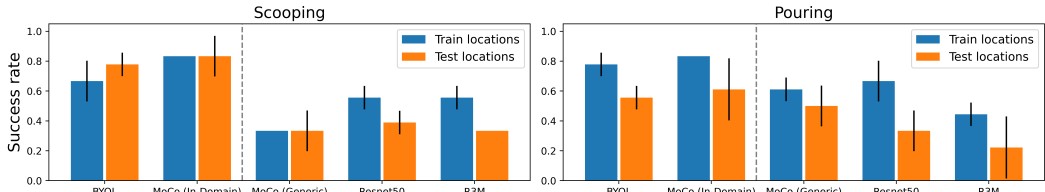

Figure 3: **Vision representation comparison with BC.** Models trained on our data (left of dashed line) perform better than generic ones (right of dashed line), and results tend to be better for training object locations than unseen test locations.

## 6.2 Policy Learning Results

Table 3 shows the comparison of policy learning methods (described in 5.2) evaluated on TOTO. Due to compute constraints, we have 1 and 2 seeds for `DT` and `IQL` respectively. We compensate by duplicating the evaluation of these seeds to keep the number of trials consistent. The results are visualized in Fig. 4. We find that `VINN` performs the best in train locations. We also note that offline-RL approaches (especially IQL) achieve some success unlike in RB2[Dasari et al., 2021]. Our dataset is larger and more diverse than RB2, likely contributing to better offline RL performance.

In experiments, we found that the scooping proves challenging due to a non-markovian aspect of the task: the spoon is above the bowl both before and after scooping. Thus we would expect open-loop methods (BC, VINN) and those with history (DT) to perform better than others in this setting. While `BC` and `VINN` achieve competitive performance on scooping, `DT` only achieves moderate success on scooping and does not see any positive rewards on pouring. Meanwhile, `IQL` provides decent performance without history on a non-markovian task.

Comparing the train and test location results for policy learning proves interesting. `VINN` performs the best on train locations but struggles on unseen locations, since it selects actions using the nearest

Table 3: TOTO policy learning results across train and test locations.

| Model | Scooping | | Pouring | |
|---|---|---|---|---|
| | Reward | Success % | Reward | Success % |
| BC + MoCo | 7.42 | **83.3%** | **22.86** | **72.2%** |
| VINN | **7.89** | 63.9% | 21.75 | 47.2% |
| IQL | 6.08 | 47.2% | 9.86 | 38.9% |
| DT | 2.83 | 27.8% | 0.00 | 0.0% |

neighbor from the training data. All other methods also experience some level of degradation when moving to unseen locations, leaving one clear direction for method improvement using TOTO.

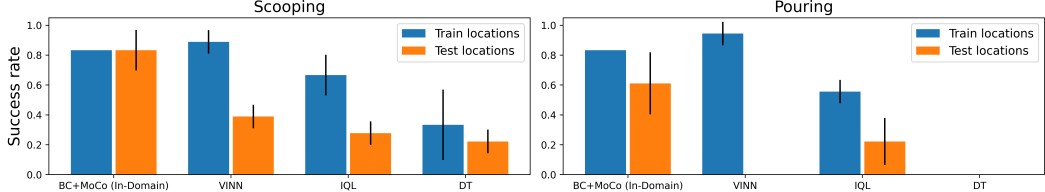

Figure 4: **Evaluating offline policy learning results.** VINN has the best performance on train locations but degrades on unseen locations, as does the performance of other methods.

## 6.3 Dataset Size Ablation

To understand the impact of dataset size on policy learning performance, we perform an ablation in which we train BC on the scooping task with varying amounts of data. We sort the scooping trajectories by reward and train policies with the top 5%, 25%, 50%, 100% of the data, as well as all the successful trajectories with positive rewards ($\sim$70%). This sorting by reward ensures that policies trained in the small-data regime are not overcome by unsuccessful trajectories. We present the dataset size ablation results in Table 4.

The *all success* number uses the same policy as the BC policy in Table 3, but we evaluate it again with the ablations to ensure minimal variance in conditions. As expected, training on more data generally leads to a higher success rate. We find that training on all data (including unsuccessful trajectories) leads to a lower reward than training on only the successful trajectories, also unsurprising given the use of BC to learn the policies in this ablation (we might expect offline RL to improve with the inclusion of unsuccessful trials).

Table 4: Dataset size ablation with BC on scooping.

| Dataset size | Reward | Success Rate |
|---|---|---|
| 5% | 2.89 | 38.9% |
| 25% | 5.94 | 72.2% |
| 50% | 6.22 | 77.8% |
| Successes ($\sim$70%) | 8.06 | 83.3% |
| 100% | 5.00 | 72.2% |

Overall, these ablation results suggest that this is the right order of magnitude for the size of the TOTO dataset in terms of policy learning. We have reached the point of diminishing returns: training on 50% versus 70% of the data does not substantially improve performance. However, additional data might still improve visual representation learning.

## 6.4 Metrics for Offline Policy Evaluation

A TOTO user might wish to sanity check their policy before submitting it for real-world evaluation or otherwise have metrics of policy performance to guide offline tuning. Here we present simple example metrics for offline evaluation: action similarity to a validation set of expert demonstrations using both joint angle error and end effector pose error. From a chosen validation set of 100 trajectories, we estimate the joint angle error and end effector error by computing the mean squared error between agent's predicted actions and actual actions for all samples.

Fig. 5 shows these validation metrics on BC checkpoints throughout training and the real-world reward evaluated on four representative checkpoints. The reward increases as the validation error metrics decrease, matching expectations. These metrics capture overfitting: the overtrained policy from 2,000 epochs shows a significant decrease in real-world reward and likewise has higher validation error. While offline metrics alone should not fully guide the development of an algorithm, it provides a signal as to whether the policy might achieve any success in the real world.

## 7 Discussion

The main goal of this work is to introduce TOTO, our robotics benchmark. We presented a broad initial set of baselines containing both vision representations and policy learning approaches, which can be built off of by future TOTO users. Notably, these baselines were contributed in the same way that TOTO will be used in the future: by collaborators who locally train policies and submit them for remote evaluation on shared hardware. This shows the feasibility of our user workflow. The initial baseline results show the challenging nature of our tasks, especially with respect to generalization. By using TOTO as a community, we can more quickly iterate on ideas and make progress on the real-world bottlenecks to robot learning.

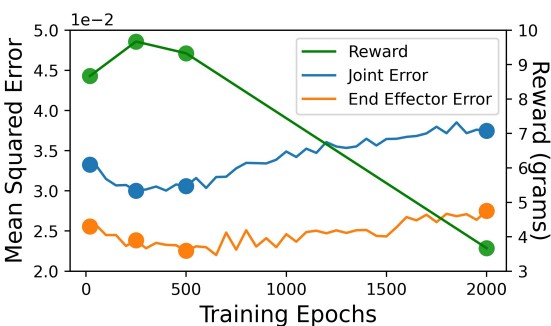

Figure 5: **Comparing offline evaluation to online performance.** While offline evaluation is imperfect, it provides a sanity check to the user, guiding development at a higher frequency than real-world evaluation.

### 7.1 Limitations and Future Work

The evaluation protocol currently has manual steps: we measure the material transferred during pouring and scooping to compute rewards and reset by returning the material to the original object. We do see future potential to automate reward measurements and resets, such as by adding a scale beneath the target object and using an additional robot to reset the transferred materials. Spills of the transferred material, however, might still require manual intervention.

We plan to expand the evaluation setup to include additional robots. This would help us meet the increasing demand in evaluations as more users adopt the benchmark. One challenge will be visual differences across robots, but we plan to collect additional demonstrations on new robots, and this would be an opportunity to expand the set of tasks as well (we could designate one robot per task).

As user demand further grows, we will implement an evaluation job queue that prioritizes evaluation requests from different users and schedules the jobs based on the number of robots currently available.

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
