# OpenReview forum: "Train Offline, Test Online: A Real Robot Learning Benchmark"
_NeurIPS.cc/2022/Workshop/Offline_RL — Offline RL Workshop NeurIPS 2022_

### Official Review · Reviewer_vorc · 2022-10-18

**Rating:** 5
**Confidence:** 3

**Review:**

This paper proposes a new manipulation robotics benchmark where it has two parts. The first part is offline data where anyone can access the data and use the data to train an imitation learning and batch RL algorithm. The second part of this benchmark, is a shared hardware setup where it seems people can remotely access it and evaluate their methods.

This is indeed an interning effort and can potentially make research and progress in robotics more streamlined and lower the entry barrier. While I appreciate this effort, I have a couple of concerns:

1.  Per my experiences and observations, once such a benchmark is publicly released, it will be abandoned and there will be no active development or support. Note that I understand it requires resources to maintain such a benchmark in the long term, I just highlight my concerns.

2. The (anonymous) link to data or benchmark has not been included in the submission.

2. I am not quite sure if remote evaluation will be scalable as it comes with lots of operational challenges and requires constant maintenance. The paper didn't discuss this in detail. It might be a better idea to use a simulator that can be used for evaluation.

This paper is a better fit for NeurIPS data and benchmark track. That said, it is not irrelevant to this workshop either.

---

### Official Review · Reviewer_kCyZ · 2022-10-20

**Rating:** 6
**Confidence:** 4

**Review:**

This paper presents a real-world benchmark for offline RL. The benchmark provides pre-collected offline data and a real-world Franka environment to evaluate the trained policy. Several BC and Offline RL baselines are evaluated on the proposed benchmark.

Strengths:

- Providing a benchmark with real-world evaluation can potentially make a high impact in offline reinforcement learning.

- Detailed data collection and evaluation protocols are presented in the paper.

Weaknesses:

- The observation space, action space, and success conditions are missing in the paper. Without such information, it is hard to justify the difficulties of the tasks.

- Only four methods are evaluated. It would be better to compare more baselines such as AWAC and CQL.

- It would be even better to provide an evaluation protocol for offline pre-training + online fine-tuning on the proposed benchmark.

- Evaluation across 3 seeds is a bit too few. 5 seeds or more would be better.

- The link (https://rb2.info/) attached in the paper does not work.